# Network dynamics of momentary affect states and future course of psychopathology in adolescents

Anna Kuranova[1]*, Johanna T. W. Wigman[1,2], Claudia Menne-Lothmann[3], Jeroen Decoster[4], Ruud van Winkel[5,6], Philippe Delespaul[3,7], Marjan Drukker[3], Marc de Hert[5,6,8], Catherine Derom[9,10], Evert Thiery[11], Bart P. F. Rutten[3], Nele Jacobs[3,12], Jim van Os[3,13,14], Albertine J. Oldehinkel[1], Sanne H. Booij[1,2,15], Marieke Wichers[1]

1 University Medical Center Groningen, University Center Psychiatry (UCP) Interdisciplinary Center Psychopathology and Emotion Regulation (ICPE), University of Groningen, Groningen, The Netherlands, 2 Department of Research and Education, Friesland Mental Health Care Services, Leeuwarden, The Netherlands, 3 Department of Psychiatry and Neuropsychology, School of mental health and neuroscience (MHeNS), Maastricht University, Maastricht, The Netherlands, 4 University Psychiatric Centre Sint-Kamillus, Bierbeek, Belgium, 5 Department of Neurosciences, Center for Public Health Psychiatry, KU Leuven, Leuven, Belgium, 6 Department of Neurosciences, Center for Clinical Psychiatry, KU Leuven, Leuven, Belgium, 7 Mondriaan Mental Health Care, Heerlen, The Netherlands, 8 Antwerp Health Law and Ethics Chair–AHLEC University Antwerpen, Antwerpen, Belgium, 9 Centre of Human Genetics, University Hospital Leuven, KU Leuven, Leuven, Belgium, 10 Department of Obstetrics and Gynecology, Ghent University Hospital, Ghent University, Ghent, Belgium, 11 Department of Neurology, Ghent University Hospital, Ghent University, Ghent, Belgium, 12 Faculty of Psychology and Educational Sciences, Open University of the Netherlands, Heerlen, The Netherlands, 13 Department of Psychosis Studies, Institute of Psychiatry, King's Health Partners, King's College London, London, United Kingdom, 14 Department Psychiatry, Brain Center Rudolf Magnus, Utrecht University Medical Centre, Utrecht, The Netherlands, 15 Center for Integrative Psychiatry, Lentis, Groningen, The Netherlands

* a.kuranova@umcg.nl

## Abstract

### Background

Recent theories argue that an interplay between (i.e., network of) experiences, thoughts and affect in daily life may underlie the development of psychopathology.

### Objective

To prospectively examine whether network dynamics of everyday affect states are associated with a future course of psychopathology in adolescents at an increased risk of mental disorders.

### Methods

159 adolescents from the East-Flanders Prospective Twin Study cohort participated in the study. At baseline, their momentary affect states were assessed using the Experience Sampling Method (ESM). The course of psychopathology was operationalized as the change in the Symptom Checklist-90 sum score after 1 year. Two groups were defined: one with a stable level (n = 81) and one with an increasing level (n = 78) of SCL-symptom severity. Group-

**Data Availability Statement:** As there is a possibility to identify participants based on their clinical and experience sampling data, the datasets generated and/or analyzed during the current study

cannot be made publicly available based on European law. The study was approved by the local ethics committee (KU Leuven, Nr. B32220107766), which has also imposed the data availability restrictions. Data requests may be sent to the TWINSSCAN general contact email address "info@twinsscan.eu".

**Funding:** Acknowledgments: The East Flanders Prospective Twin Survey (EFPTS) is partly supported by the Association for Scientific Research in Multiple Births and the TwinssCan project is part of the European Community's Seventh Framework Program under grant agreement No. HEALTH-F2-2009-241909 (Project EU-GEI); M. Wichers was supported by funding from the European Research Council (ERC) under the European Union's Horizon 2020 research and innovative programme (ERC-CoG-2015; No 681466); J.T.W Wigman was supported by the Netherlands Organization for Scientific Research (NWO) (Veni grant no. 016.156.019); Ruud van Winkel was supported by a FWO Senior Clinical Fellowship (3M150375). The funders had no role in study design, data collection and analysis, decision to publish, or preparation of the manuscript. We would like to thank the Center for Information Technology of the University of Groningen for their support and for providing access to the Peregrine high performance computing cluster.

**Competing interests:** The authors have declared that no competing interests exist.

level network dynamics of momentary positive and negative affect states were compared between groups.

## Results

The group with increasing symptoms showed a stronger connections between negative affect states and their higher influence on positive states, as well as higher proneness to form 'vicious cycles', compared to the stable group. Based on permutation tests, these differences were not statistically significant.

## Conclusion

Although not statistically significant, some qualitative differences were observed between the networks of the two groups. More studies are needed to determine the value of momentary affect networks for predicting the course of psychopathology.

## Introduction

Mental disorders place a heavy burden both on individuals and society [1]. First symptoms of psychopathology often emerge during childhood and adolescence [2, 3]. For some adolescents, these symptoms persist and may develop into fully manifested mental disorder later in life. There are various risk factors of future mental illness. Some are genetic [4], whereas others are environmental, such as early life adversities and traumas [5–9]. Yet, even among individuals exposed to these risk factors, only a small proportion develops clinical levels of psychopathology, accompanied by impairment and need for care [6]. For the purpose of prevention, it is crucial to understand why some people develop more severe symptoms and others do not.

A better understanding of the underlying mechanisms of psychopathology may improve adequate identification of children and adolescents at risk for developing (severe) psychopathology. One approach to uncovering these mechanisms is zooming in to the moment-to-moment patterns of affect, experiences, and thoughts in the flow of daily life. Such dynamical patterns may be effectively assessed with experience sampling (ESM) design, i.e. collecting intensive time-series data on momentary experiences multiple times during the day [10]. Previous studies using ESM have shown that changes at the level of daily life experiences are associated both with risk factors and future development of psychopathology. Among the risk factors shown to be associated with the altered dynamics of affect states in daily life are genetic risk [11–13], certain personality traits [14], childhood adverse experiences [15–17], and poor sleep quality [18–20]. In turn, the altered dynamics of affect states in daily life have been shown to predict the emergence of new symptoms later [21–24]. Taken together, these findings suggest that the way in which momentary experiences interact with each other in daily life, i.e. the dynamics between moment-to-moment affect states, may influence the impact of risk factors on the later manifestation of psychopathology.

If part of the underlying mechanisms of psychopathology can be inferred from the daily life dynamics of affect states, then the important question arises as to how these dynamics in such short-lived experiences can substantially influence the future development of symptoms. One theory is that the change in a single affect state can set in motion a cascade of changes in other experiences and behaviors [25, 26]. For example, for some people, feeling lonely may induce states of feeling down and irritated. These affect states, in turn, may re-activate feeling lonely.

Such mutual influences, when occurring repeatedly, can lead to 'vicious cycles' of affect states that keep reinforcing each other, trapping a person in a negative flow. Yet, for others, feeling lonely may pass without activating other negative affect states, or may be neutralized by a later positive affect state (e.g. feeling cheerful after seeking for social support from peers). Moreover, the ability of positive states to interrupt or downregulate the negative "vicious cycles" may be associated with resilience to psychopathology and may represent an important part of its mechanism. Thus, the impact of a minor mood perturbation may vary depending on the dynamics of affect states. To investigate these dynamics, we need to assess the whole system of interacting positive and negative affect states.

These ideas align with the network theory of psychopathology [27, 28]. According to this theory, symptoms of mental disorders may emerge not due to some "hidden", underlying cause, but due to direct interaction with each other. For example, insomnia may influence performance at school and peer relationships, and through that increase rumination and lower self-esteem. These effects can be visualized as a network of interacting symptoms or states, and analyzed as a whole system as well as individual elements. Empirical support for this approach is growing. Several recent studies suggest that negative affect states influence each other more strongly and might have a higher tendency to form 'vicious cycles' in individuals with psychopathology compared to healthy controls [25, 29–32] although other studies found mixed results [33, 34] or did not find this effect [35, 36]. However, because most of these studies compared patients or high-risk individuals with healthy controls (or patients with high and low level of symptoms [36]), it is possible that the observed differences in affect dynamics between these groups are the result of already developed psychopathology, rather than be the cause of it. To determine whether characteristics of the dynamics between momentary affect states are key factors in the developmental process of symptom formation, we need to examine whether these characteristics are already present in populations at increased risk for psychopathology, before more severe symptoms arise. The reasoning behind including individuals at increased risk is that any underlying vulnerability for, as well as resilience against, psychopathology can be exposed only when challenged by risk factors. Because (i) adolescence is a sensitive period for the development of psychopathology in which symptoms often emerge for the first time [37, 38], and (ii) a low level of happy childhood experiences is a known risk factor for psychopathology [39, 40], adolescents with low levels of happy childhood experiences represent a well-suited population for this purpose.

Therefore, we aim in this paper to explore whether the dynamic network structure of affect states differs between adolescents who develop higher level of symptoms over time and adolescents with a relatively stable level of symptoms. We used a prospective research design in an adolescent population with experience sample (ESM) data collection carried out at baseline and with follow-up assessments to differentiate the course of future psychopathology. We hypothesize that affect state networks of individuals who are vulnerable to the development of future psychopathology will show dynamics of affect states that are prone to the development of vicious cycles. For such individuals, negative affect states will have strong mutually reinforcing connections. Furthermore, we hypothesize that in networks of individuals who are resilient against psychopathology (i.e. do not develop new or more severe symptoms despite being at an increased risk), positive affect states have the potential to interfere with such vicious cycles by down-regulating one or more of these negative affect states. Stated specifically in terms of network characteristics, we expect that the network of affect states in adolescents with a future increase in the level of symptoms compared to the network of affect states of adolescents with a relatively stable symptom level (i) contains stronger connections between negative affect states, (ii) contains positive affect states that are less influential in the network, and (iii) has a dynamical structure between affect states that predisposes to vicious cycles.

## Methods

### Sample and design

Data were obtained from the longitudinal prospective study 'TWINSSCAN' ("http://www.twinsscan.eu"; website only in Dutch), a cohort nested in the East-Flanders Prospective Twin Study (EFPTS), a register of all multiple births in the Province of East Flanders, Belgium, from 1964 onwards [41, 42]. In 2010 potential participants for the TWINSSCAN cohort were recruited by sending invitation letters to all EFPTS participants who were between the ages of 15 and 18 years. Furthermore, to recruit more twins and their non-twin siblings between ages of 15 and 34, a general invitation was included in a newsletter from the EFPTS. All participants provided their written informed consent. For those participants who were aged below 18 years, their parents or caretakers provided additional written consent. The local ethics committee (KU Leuven, Nr. B32220107766) approved the study.

The TWINSSCAN sample enrolled in the baseline assessment comprised 839 people and involved a broad range of measurements, including clinical interviews, questionnaires, experiments and an ESM period [43]. For the current paper, additional inclusion criteria applied based on available data. First, participants needed to score below the median on items assessing the happiness of their childhood (see Measures for more details), leading to the exclusion of 388 individuals. Second, they needed complete data on the Symptom Check List-90 (SCL-90) [44] at both baseline (T0) and a follow-up wave after one year (T1), leading to the exclusion of another 202 individuals. Third, we excluded 10 individuals with more than 30% missing ESM data points. Altogether, this resulted in a sample of 239 participants, who were grouped according to their pattern of SCL-90 symptom change over one year (see details below). This change score was divided into tertiles, representing groups with decreasing, stable and increasing levels of symptoms. The group with decreasing symptoms was excluded for theoretical reasons (see details below), resulting in a final sample of 159 individuals, categorized into a group with Stable symptom levels (n = 81) and a group with Increasing symptom levels (n = 78).

### Measurements

**Quality of childhood experiences.** As our research question can best be examined in a sample at risk for psychopathology, we used four items of the Dutch questionnaire on adverse childhood experiences (JTV) [45] to assess the quality of childhood experiences, namely the items: 'I had a happy childhood', 'my parents greatly loved each other', 'I got the attention that I needed', and 'my privacy was respected'. These four items were over 90% correlated with the overall score of the JTV questionnaire that was used in a previous twin sample of the EFPTS (see Jacobs et al. [46], for a description of this sample). In addition, they showed optimal variation in the studied population, as they are phrased positively. Therefore, for the current data collection, it was decided to assess only these four items, as it relieves the participants' burden of filling out questionnaires, but retains essential information. These items were measured with five points Likert scale ranging from 1 ("Never") to 5 ("Very often").). These four items had good internal consistency (Cronbach alpha for these four items in our sample was 0.83 (Confidence Interval: 0.80–0.85)). The sum score of the four items was calculated, and a median split of the sum score of the four items was used to define a high-scoring and low-scoring individuals on safe and happy childhood experiences, of whom the former were excluded from further analysis (see 'Sample and design')."

**Subclinical psychopathology.** The presence of general psychopathological symptoms was assessed using the Symptom Check List-90 (SCL-90) [44]. The items assess the level of distress associated with general and specific psychopathological symptoms with 5 points Likert scale

ranging from 1 ("Not At All") to 5 ("Extremely"). Following previous research suggesting that all 90 items measure one common construct of psychopathological problems [47], a sum score of all 90 items was used in the analysis. All participants in the final sample (see Results for detailed description) completed all 90 items.

**Group composition.** To assess change in the level of symptoms, we subtracted the SCL-90 scores at T0 from the SCL-90 scores at T1 for each participant. After that, these change scores were divided into tertiles, resulting in 3 groups defined by a reduction (Decrease group, mean SCL-90 sum score change = -41.48 points, SD = 33.09, n = 80;), minimal change (Stable group, mean SCL-90 sum score change = -5.02 points, SD = 4.95, n = 81) and an increase in symptom level (Increase group, mean SCL-90 sum score change = 25.66, SD = 22.5, n = 78) (see also Table 1). As the group with a future symptom reduction, the Decrease group (tertile 1) reported significantly higher scores on the SCL-90 at baseline than the other two groups (see Results for details), we excluded the Decrease group. The reason for exclusion is that when comparing networks of groups of people with different levels of symptoms, we cannot eliminate the possibility that the differences in estimated network paths are explained by differences in variances between the groups in ESM items [48]. Therefore, this group could not be used to test the current hypothesis. Hence, we analyzed data from the Stable and the Increase groups only, leaving 159 individuals for the final analysis (for details see Results section).

**The experience sampling method.** In this study, participants received a custom-made PsyMate[tm] device, developed for the specific purpose of collecting ESM data (www.psymate. eu). For six days, participants completed short questionnaires (around 40 items, with additional items on mornings and evenings) about their current affect states, thoughts, daily life context and behavior. The devices were programmed to beep ten times a day at semi-random moments between 07:30 am and 10:30 pm, with the 90 minutes between beeps on average. Participants were instructed to fill out the diaries immediately after the beep. Only observations with all present ESM items were included in the analysis. Similar to previous studies, we have excluded participants with more than 30% missing observations [11, 49]. More details regarding the procedure of ESM methodology can be found elsewhere [10, 21].

**ESM measures.** We selected ESM items based on both theoretical and methodological criteria. First, we only selected experiential affect states (not thoughts, behaviors or context information). Second, of these, we selected at least one item from each quadrant between the axes of "pleasure" and "arousal" as defined in the circumplex model of affect [50, 51]. Additionally, we added the items "Down", and an item "Energetic", as they reflect common transdiagnostic symptoms [52, 53]. Third, to avoid a floor effect because of insufficient variance [48], we chose items with a within-person standard deviation (SD) of around 1.0 (see Table 1). Fourth, we chose affect states that were not highly correlated with each other (r < 0.5), so that all items captured different aspects of a momentary mental experience. Fifth, to ensure that the differences between group networks originated from differences in the dynamics between affect states, we checked whether the mean levels of the selected items did not differ between the Increase and Stable groups, and whether the within-person SDs of the selected items did not differ more than 10–12%. As a result, we included the following six affect states: 'cheerful,' 'relaxed,' 'energetic,' 'irritated,' 'down,' and 'lonely.' The items were formulated as follows: 'At this moment I feel. . . ('Down', for example)'. The items were assessed with 7 points Likert scales from 1 ('not at all') to 7 ('very much').

## Analysis

We sought to investigate the dynamic interrelations between affect states and visualize those interrelations as networks of affect states for each group. The ESM data had a multilevel

**Table 1. Sociodemographic characteristics, level of happy childhood experiences (JTV), Symptom Check List-90 scores, and mean levels and SDs of ESM variables for the Stable and Increase groups.**

| Measure | | The Stable group | | | The Increase group | | |
|---|---|---|---|---|---|---|---|
| Number of people | | 81 | | | 78 | | |
| % and n females | | 69.14% (56) | | | 62.82% (49) | | |
| % and n education | Low education | 9.88% (8) | | | 5.13% (4) | | |
| | Middle education | 61.73% (50) | | | 70.51% (55) | | |
| | High education | 28.40% (23) | | | 21.79% (17) | | |
| | No data | 0.00% | | | 2.56% (2) | | |
| Ethnicity | Caucasian | 79 | | | 77 | | |
| | Asian | 1 | | | 0 | | |
| | No data | 1 | | | 1 | | |
| | | *M* | *SD* | *Range* | *M* | *SD* | *Range* |
| Age | | 17.86 | 3.96 | 14–33 | 16.92 | 3.58 | 15–34 |
| JTV scores* | | 15.58 | 1.56 | 11–17 | 14.95 | 2.14 | 7–17 |
| SCL-90 at baseline | | 126.8 | 26.1 | 92–214 | 130.24 | 34.0 | 90–245 |
| SCL-90 change | | -5.04 | 4.95 | -13 - +4 | +25.7 | 22.5 | +5 - +105 |
| SCL-90 at the follow-up* | | 121.78 | 25.8 | 90–212 | 155.90 | 42.4 | 98–305 |
| Number of filled-in ESM observations | | 43.4 | 10.5 | 22–76 | 41.7 | 11.2 | 20–79 |
| Number of filled-in 2 consecutive ESM observations | | 32.4 | 12.1 | 11–67 | 31.0 | 13.6 | 7–76 |
| | | *M* | *SD within-person* | | *M* | *SD within-person* | |
| Cheerful | | 4.76 | 1.16 | | 4.53 | 1.29 | |
| Relaxed | | 5.03 | 1.15 | | 4.86 | 1.26 | |
| Energetic | | 4.63 | 1.09 | | 4.34 | 1.14 | |
| Irritated | | 2.24 | 1.20 | | 2.41 | 1.34 | |
| Down | | 1.79 | .96 | | 1.91 | .96 | |
| Lonely | | 1.69 | 1.04 | | 1.86 | 1.09 | |

Note: * Corresponds to a significant difference (<0.05) between Stable and Increase groups.

JTV is 4 items ('I had a happy childhood', 'my parents greatly loved each other', 'I got the attention that I needed', and 'my privacy was respected') from Dutch questionnaire on adverse childhood experiences, with higher scores reflecting higher level of happy childhood experiences (Arntz et al., 1996). SCL-90 is from Symptom Check List-90 (SCL-90) questionnaire (Derogatis, 1977), sumscore of all items.

structure (multiple observations (level 1) within one person (level 2), and multiple persons within a twin pair (level 3)). Therefore, we used autoregressive multilevel linear models to test how each affect state (e.g. 'cheerful') at each time point (t) was predicted by itself and all other affect states at the previous time point (t-1) (see the Fig 1 for the regression equation).

Before the modelling, we person-mean centered the selected ESM items in order to keep only the within-person effects in the models. We chose not to standardize as obvious reasons to standardize did not apply. All the variables were on the same 7-point Likert scale with similar anchors, and we selected items with similar variance [54, 55]. Hence, b-coefficients from the models can be straightforwardly compared across individuals and ESM variables. Furthermore, beep lags over the night were excluded. All analyses were conducted in R with the 'nlme' package [56], see S1 Data for R script.

The models were fitted separately for the Stable and the Increase group. Resulting b-coefficients were used as values representing the effects of affect states on each other, and those values were used to construct dynamical networks for the two groups.

In the multilevel models, a separate variable representing time (the beep number over the whole ESM period) was added in order to account for possible trends over time. For the

**The example equation for the model of "Cheerful":**

$Cheerful_{ijk} = (\beta_0 + e_{ijk}) + (\beta_1 + u_{1ijk}) * Cheerful_{ijk}{}^{-lag} + (\beta_2 + u_{2ijk}) * Relaxed_{ijk}{}^{-lag} + (\beta_3 + u_{3ijk}) * Energetic_{ijk}{}^{-lag} + (\beta_4 + u_{4ijk}) * Irritated_{ijk}{}^{-lag} + (\beta_5 + u_{5ijk}) * Down_{ijk}{}^{-lag} + (\beta_6 + u_{6ijk}) * Lonely_{ijk}{}^{-lag} + (\beta_7 + u_{7ijk}) * Time_{ijk};$

With $\beta_0$ being an intercept, $\beta_1 - \beta_7$ being regression coefficients, i corresponds to the level of assessments, j for the person level, k for twin level, $u_{1ijk} - U_{7ijk}$ stands for the random slopes, and $e_{ijk}$ for the error.

**Fig 1.**

random effects on both the individual and twin level, we added a random intercept. On the level of individuals, we also added random slopes for time and for all ESM variables in order to correct for individual differences in trends in these variables. For the random effects covariance matrix, we used a diagonal structure [56]. For the residual covariance matrix, we used a continuous AR(1) correlation structure [56]. Both structures were chosen based on the possibility for the model convergence and the best model fit based on AIC comparison. For all 12 models, the assumptions of normality of the residuals distributions were checked with visual inspection.

## Networks of affect states

Each ESM affect state variable was depicted as an individual node in the network and the b-coefficients of the fixed effects (i.e. the time-lagged effects in the six multilevel models) represented directed connections (edges) in the networks. The networks were visualized using the 'qgraph' R package [57]. For a more straightforward visual comparison of the networks, the maximum strength of the connections was set equal in both networks; this insured the match between the thickness of the edge to the same numeric value for both networks [57].

## Comparison of group networks

We used both descriptive assessment and a permutation testing procedure (by W. Viechtbauer [25]) for the statistical comparison of the networks. The idea behind the permutation approach is to randomly combine outcomes with predictors in repeated permutations (here 10000 times) and test the probability of obtaining the same results seen in the actual data (for details see S1 Text). Permutation tests in dynamic networks have been scarcely applied [25, 34, 58]. Also, the precise power the procedure needs to discriminate effects in dynamic networks is still unknown. One study simulated power for this but only for cross-sectional data networks, and not for dynamic ones [59]. Therefore, we decided to consider both descriptive network outcomes, similar to previous network studies [25, 29–32, 34] as well as statistic network outcomes.

For the first aim, to investigate whether the network of affect states of individuals who will develop more symptoms over time (Increase group) contained stronger connections between negative affect states compared to network of affect states of individuals with the relatively stable level of symptoms (Stable group), we used a permutation test to quantitatively compare the strength of the connections between negative affect states for the Stable and the Increase groups. To this end, we calculated and compared sums of all absolute b-coefficients from the regression models for all the paths between negative nodes. All estimated network paths were chosen based on the common practice in the field to use all available information [32–34]. For

the second aim, to investigate whether positive affect states are less influential in the networks of affect states in the Increase group compared to the Stable group, we compared the two groups in terms of the influence of the positive affect states. To do that, we used a permutation test to quantitatively compare (i)) the relative importance of the positive nodes in the networks, based on their out-strength centrality measures; (ii) the overall effect of the positive states ('cheerful', 'relaxed', 'energetic') on the negative states ('irritated', 'down', 'lonely') and vice-versa. Out-strength centrality measure is a network characteristic that equals the sum of all connections going from the node of interest to the other nodes, and reflects the overall influence of this node on the other ones. Specifically, the out-strength centrality was calculated by summing the b-coefficients from the regression models for the indicated paths. All of these differences were compared both descriptively and with the permutation test. For the third aim, to investigate whether the network of affect states of the Increase group has a dynamical structure that predisposes more strongly to vicious cycles than of the Stable group, the networks of two groups were qualitatively compared, based on visual inspection (without using a permutation test). For this aim, only significant paths ($p < .05$) were visualized and considered, as visual inspection of all available paths is not informative. Moreover, to ensure the robustness of the results of the visual inspection, we performed a limited version of multiverse analysis (based on [60]) to test the influence of different group compositions based on different cutoffs for the SCL-90 change score. A detailed explanation of the calculations, the visualization, the assessment and the limited multiverse analysis can be found in the supplementary materials (S1 and S2 Texts and S3 Table and S1 Fig).

## Results

### Groups

The final sample (n = 239) was grouped based on tertiles of change in their psychopathological trajectory over the course of one year. This led to three groups: a Stable group (n = 81) with a relatively small decrease in symptoms (for details see Table 1);, an Increase group (n = 78) with a relatively large increase in symptoms (for details see Table 1), and a Decrease group (n = 80), with a relatively large decrease in symptoms ($M_{age}$ = 17.84, age range: 14–33 years, SD = 3.84; 66.25% females). As the latter subgroup had significantly ($p < .0001$) higher SCL-90 scores at baseline (mean level 168.3, corresponding to "high" symptom level in the normal population [61]) than the other two groups, i this group was excluded from analyses. The Stable and the Increase group did not differ significantly on the baseline SCL-90 score (mean level of SCL-90 for the Stable group = 126.8, for the Increase group = 130.24, difference = 3.44, p = .48), and their levels correspond to "mean"/"above mean" levels in a normal population [61]. At T1, the level of symptoms of the Increase group was equal 155.90 (corresponding to "high" levels in the normal population [61]) and significantly higher than that of the Stable group (mean level 121.78) with difference = 34.13, p<0.001 which roughly corresponds to an increase of one severity category [61]. Trajectories of psychopathology for the two groups are presented in Fig 2.

The Stable and Increase groups did not significantly differ in socio-demographic characteristics and mean levels of each ESM variable at T0 (Table 1). Ratios of within-person variances for all ESM variables did not differ more than by 10.1% between groups. The Increase group had significantly lower level of happy childhood experiences (JTV) (difference = 0.63, p = 0.04)

### Dynamic affect networks

The networks of affect states for the Stable and Increase groups are presented in Fig 3 (significant paths) and S1 Fig (all paths). S1 Table shows a table of the b-coefficients of the time-lagged

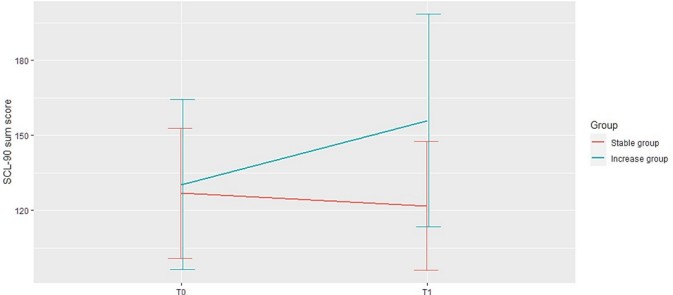

**Fig 2. The change in SCL-90 mean sum score for the Stable and the Increase groups.** In this figure, the y-axis represents the total sum score of the SCL-90 items; x-axis represents the baseline (T0) and one year (T1) assessments. The lines represent the change in the mean number of symptoms for the Stable group (lower red line) and for the Increase group (upper blue line). Vertical lines represent the standard deviations of the mean SCL-90 scores for the Stable group (a red line slightly to the left) and the Increase group (a blue line slightly to the right) on T0 and T1. The Stable and the Increasing group did not differ significantly on the SCL-90 score (difference = 3.44, p = .48) at T0 At T1, the level of symptoms of the Increase group was significantly higher than of the Stable group with difference = 34.13 (p<0.001), which roughly corresponds to the differences in severity categories between «above middle» and «high» (Arrindell et al., 2003)).

effects from multilevel models. For all models, assumptions of normality of the residuals distributions were met.

**Comparison of the group networks.** For the first aim, we compared the total strength of the network connections between negative affect states (negative connectivity) for the Stable and the Increase groups. The network of the Increase group had a more than twice as high level of connections between negative affect states (.29) than the network of the Stable group (.13). This difference (.17; 229%) was not confirmed statistically (S2 Table).

For the second aim, we compared the influence of positive affect states in the networks between the Stable and the Increase group. First, we compared the relative importance of the positive nodes in the networks, based on their out-strength centrality measures. The largest difference was found for the node 'cheerful', with a higher value in the Stable group (.28 in the Stable group, .18 in the Increase group, difference = .10, 158%). For 'energetic', the relative importance of this node was higher in the Increase group (.24 in the Stable group, .36 in the Increase group, difference = .12, 148%), and for 'relaxed', values were almost similar for the

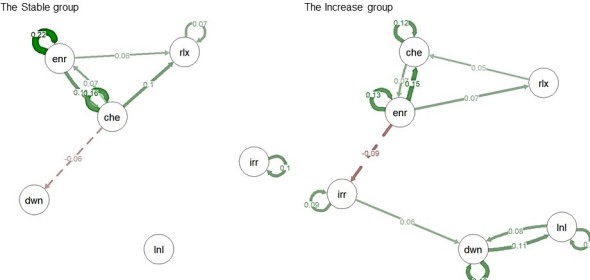

**Fig 3. Networks of affect states: Significant paths.** In this figure, affect states networks are visualized for the Stable and the Increase groups. Only significant paths (p<0.05) are presented. Presented are temporal networks, meaning that the connections represent the effect of the variable at time point t-1 on the variable at the time point t. Solid green edges represent positive connections from one node to the other, meaning that the increase in one node variable at time point t-1 is associated with increase in the other variable at time t. Dashed red edges represent negative connections, meaning that the decrease in one node variable at time point t-1 is associated with decrease in the other variable at time t. Circular edges represent autocorrelations, i.e. the effect of the variable at time point t-1 on itself at t. 'Che' -'cheerful', 'rlx' -'relaxed', 'enr' -'energetic', 'dwn' -'down', 'irr' -'irritated', 'lnl'—'lonely'.

two groups (.12 in the Stable group, .10 in the Increase group, difference = .02, 124%). None of the above differences were significant (S2 Table).

Second, we compared the overall effect of the positive states ('cheerful', 'relaxed', 'energetic') on the negative states ('irritated', 'down', 'lonely') and vice-versa (Fig 4). We found that the positive affect states were more strongly associated with lower subsequent levels of negative ones in the Stable group than in the Increase group (For the Stable Group = .21, for the Increase group = .16, difference = .06, 136%). Negative affect states were more strongly associated with lower subsequent levels of positive affect states in the Increase group (For the Stable Group = .13, for the Increase group = .21, difference = .09, 166%). However, these differences were again not significant according to the permutation test (S2 Table).

Lastly, for the third aim, we compared the networks of the affect states for the Stable and the Increase groups descriptively for the presence of dynamical structures that can predispose to vicious cycles. Visual inspection of the networks revealed qualitative similarities and differences in the structure of the networks between the groups (see Fig 3). We observed in the networks of both groups similar positive (covering nodes 'cheerful', 'relaxed', 'energetic') clusters of interconnected nodes. Furthermore, in both groups, the positive cluster had a connection associated with a subsequent reduction in the negative nodes, namely to node 'Down' in the Stable group, and to node 'Irritated' in the Increase group. However, the Increase group had an additional cluster of interconnected negative nodes (covering nodes 'down', 'lonely', and 'irritated'), with bidirectional paths between 'Down' and 'Lonely'. The negative nodes in the Stable group, however, were not connected and could therefore not form a vicious cycle. These networks differences were robust to the changes in group allocations based on the limited multiverse analysis (see S2 Text and S3 Table for details).

## Discussion

The purpose of this study was to investigate whether the presence of differences in the dynamic networks of momentary affect states precedes the development of more severe psychopathological symptoms in adolescents at an increased risk. In this study, we examined, both statistically and descriptively, whether differences in the dynamical networks of affect states at baseline can be found between groups of adolescents with increasing and relatively stable levels of psychopathological symptoms over one year. For aims one and two of the study, although

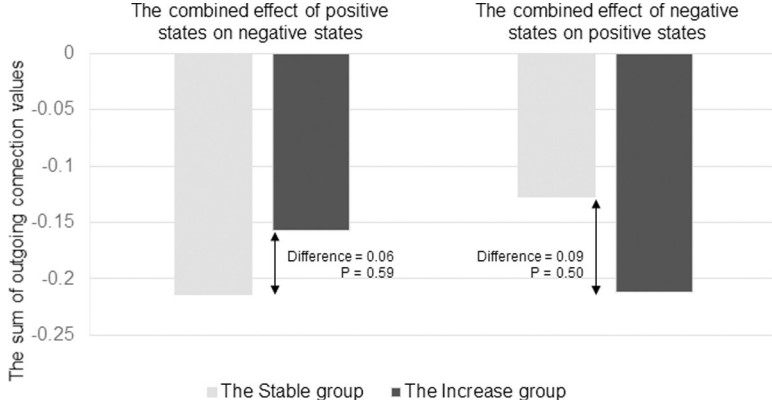

**Fig 4. The combined effect of the positive states on the negative states and vice-versa for the Stable and the Increase groups.** In this figure, the y-axis represents the summed value of all outgoing connections from the positive affect stated towards negative ones, and vice-versa. The light-grey bar represents the effect for the Stage group, and the black bar the Increase group. Depicted differences between groups are not significant according to the permutation test.

all differences were consistent with the hypothesized expectations, the observed quantitative differences were not confirmed by statistical tests. For the third aim of the study that was assessed descriptively, results cautiously suggest that, compared to adolescents who appeared resilient against psychopathology, the dynamic structure of affect states in adolescents who developed more severe psychopathology over time had a dynamical structure between affect states that could predispose to vicious cycles.

## Comparison to previous studies

The null results for the quantitative aims in our study are partly inconsistent with several previous studies that suggests the existence of both quantitative and qualitative differences between networks of people with and without psychopathology or risk of it [11, 29–31, 34, 62]. There are several explanations for this discrepancy. First, most of the previous studies compared the networks of individuals based on *current* levels of psychopathology and did not focus on the development of future psychopathology. Some other studies comparing groups based on the follow-up measures also did not find any differences [36] or found mixed results [34]. In addition, none of these studies can be directly compared to the current one, as they either compared patients with healthy controls [25, 29–31], or followed patients with MDD through the course of recovery and treatment [34, 36]. In our study, however, the dynamics of affect states were assessed a year before the new symptoms arose. Hence, the observed quantitative differences at this stage might be too small to be detected by permutation testing, as the differences in effects were quite large (~160% on average) and almost all effects were in the expected direction. This suggests that there may be quantitative differences, but that the permutation tests were too power-hungry to detect them. Moreover, regarding these effect sizes and their relevance, it is yet unknown how large differences in network dynamics need to be to have the potential to influence the future course of symptoms. It is a well-known characteristic of complex systems that even the slightest perturbation may lead to dramatic differences over time [63]. Therefore, it is possible that the observed effects are enough to be relevant over time, even though they are not statistically significant at the moment of assessment.

Other, more methodological, explanations of the null findings in this study (opposed to most previous studies) may relate to the fact that some of the previous studies did not use any statistical tests and showed different mean levels of ESM affect states, which can lead to different variances between groups because of floor and ceiling effects. Therefore, contrasting levels of means and variances of ESM items in previous studies may have contributed to artificial differences in connection strengths [48], and hence, may have led to the disparate networks and unrealistically large effect sizes. In the current study, we tackled this problem by using groups of individuals with similar levels means and variances of ESM variables.

Considering the qualitative findings, they were in line with the theoretical expectations and previous research [25, 26]. A main advantage of the network approach is the possibility to investigate the pathways in a network, i.e. which variables are connected to which ones, in which directions and how. To compare such pathways within networks, no statistical test currently exists, and yet these pathways may contain essential information about the mechanisms of psychopathology. Therefore, we argue that even in the absence of quantitative differences between networks, the qualitative differences (based on the visual assessment) may be important to take into account.

## Qualitative results: Negative cluster and possibilities for 'vicious cycles'

Based on visual assessment, the most striking visual difference between groups was the absence of the negative affect cluster in the network of the Stable group, because the negative items

were not connected to each other in this group. In the Increase group, on the contrary, the negative items were interconnected, and there was a loop of reinforcing connections ('vicious cycle') between the items "Down" and "Lonely". If negative affect states can easily trigger other negative states, then a vicious cycle may ensue that, by the accumulation of these small effects, may contribute to the development of symptoms. In the current results we see that a network structure that could facilitate the emergence of such 'vicious cycles' between affect states was present only in the Increase group. Thus, these findings partly aligns with theory suggesting that "vicious cycles" and high clustering between negative states may be present in psychopathology [26, 64, 65]. Furthermore, these findings further support previous results on differences in network structures between depressed and healthy people, which also showed more connectivity between negative affect states in the depressed groups than in the healthy groups [29, 30]. In the current study, however, in contrast to the previous studies, we can be sure that differences in the structure of the network were not the result of baseline differences in symptom levels. Thereby, this study adds important information to this field.

## Methodological issues

There are several limitations to our study. First, the sample used had several features that limit generalizablility of the findings: (I) the data came from a twin sample, and the dynamics of affect states may have a shared hereditary component. However, we could only use those participants who also had follow-up measurements which lead to a sample size that was too small to address this.; (II) the sample consisted almost exclusively of Caucasian participants, which limits the generalizability of the finding to other populations, and (III) although most participants were adolescents (mean age = 17.46), emerging adults were also included in the TWINSSCAN cohort. We decided to keep their data in the analysis because we were interested in the sensitive period for the development for psychopathology, which is broader than adolescence per se [37, 66]. Moreover, keeping these participants slightly increased the power of our study and its consistency with other studies using the same dataset.

Second, we made several methodological decisions that may have impacted the results. (I) the sample was created by selecting the 50% of people with the lowest level of happy childhood experiences, and the SCL-90 change scores were split into tertiles. Although these decisions are, to a certain extent, arbitrary, they were based on theoretical (e.g., interest in those at highest risk) as well as methodological (e.g., optimising subgroup size) reasons. In addition, the results were robust to changes in group allocations, supporting our confidence in the choices made. The limited multiverse analysis that we ran to investigate the potential effect of our subgroup selection strengthens us in our assumption that our choice was solid (see S2 Text and S3 Table). (II) Due to the rigorous selection of the ESM items, it is possible that some affective experiences that play an important role in the network were not included. To minimize this possibility, we included affect states out of all four quadrants of the affect grid from the model of Barrett & Russell (1998) [50]. (III) we used the total score of the SCL-90 as an indicator of general psychopathological severity; this could also have led to averaging out any changes in specific areas (e.g., depression). However, using a general index is in line with current views in the field of psychopathology as a broad, transdiagnostic or even one general factor [67, 68]. In addition, because this sample is at risk but not diagnosed for any particular psychiatric disorder), we feel that the use of a general index is most suitable. (IV) A threshold of a (around) 10% difference in within-person standard deviations of ESM variables between groups was conservative, but arbitrary. Therefore, we cannot completely rule out that the differences between groups are partly explained by the remaining differences in the items variances. However, as this threshold is conservative and we chose items with a considerable amount of

variation, it is unlikely that floor effects were present in the data that could have biased the findings. (V) For the random effects covariance structure the diagonal matrixes were used, as models with more complex structures did not converge. Therefore, although we modelled random slopes for all affect variables, thus captured individual differences in the changes of these affect variables through time, the correlations between these differences were not taken into account.

Third, there were several features of the study design that also could have influenced the results. The course of psychopathology was assessed only at two time points, at baseline and one year after, and therefore only a part of the full developmental trajectory is captured. Finally, this study was conducted at the group level, as we examined average differences in the structures of group dynamic networks, and therefore it is not possible to directly estimate the effects for individual cases. This would be an important next step. Identification of dynamical patterns at the individual level and connecting those to future changes in symptoms will add new evidence on the relevance of the network approach to affect dynamics and may yield promising targets for future personalized diagnostic, prevention and treatment strategies [69–71].

Taken together, our findings cautiously suggest that some differences in dynamical networks of affect states of adolescents with different mental health trajectories may exist already one year before new symptoms develop. However, these differences may be subtle and not yet statistically detectable by the permutation testing approach. Hence, more studies examining these qualitative indicators at an early stage are needed to give a more definite answer as to whether these emotion dynamics can be detected in a reliable way, and if so, how they may be used to create new methods of treatment and prevention of psychopathology.

## Supporting information

**S1 Table. The network connections between ESM variables based on the B-coefficients from the autoregressive multilevel regression models and 95% confidence intervals for B-coefficients.**
(DOCX)

**S2 Table. Results of the permutation test.**
(DOCX)

**S3 Table. All possible networks of "stable" and "increase groups" and characteristics of the "stable" groups.**
(DOCX)

**S1 Text. The detailed explanation of the calculations for the aims of the study.**
(DOCX)

**S2 Text. Limited multiverse analysis.**
(DOCX)

**S1 Data. R script.**
(R)

**S1 Fig. Networks of affect states: All paths.** In this figure, affect states networks are visualized for the Stable and the Increase groups. All paths are presented without considering their statistical significance. Presented are temporal networks, meaning that the connections represent the effect of the variable at time point t-1 on the variable at the time point t. Solid green edges represent positive connections from one node to the other, meaning that the increase in one node variable at time point t-1 is associated with increase in the other variable at time t.

Dashed red edges represent negative connections, meaning that the decrease in one node variable at time point t-1 is associated with decrease in the other variable at time t. Circular edges represent autocorrelations, i.e. the effect of the variable at time point t-l on itself at t. 'Che'—'-cheerful', 'rlx' -'relaxed, 'enr' -'energetic', 'dwn' -'down', 'irr' -'irritated', 'lnl' -lonely'.
(TIF)

## Author Contributions

**Conceptualization:** Anna Kuranova, Johanna T. W. Wigman, Albertine J. Oldehinkel, Sanne H. Booij, Marieke Wichers.

**Data curation:** Claudia Menne-Lothmann, Jeroen Decoster, Ruud van Winkel, Philippe Delespaul, Marjan Drukker, Marc de Hert, Catherine Derom, Evert Thiery, Bart P. F. Rutten, Nele Jacobs, Jim van Os, Marieke Wichers.

**Formal analysis:** Anna Kuranova, Marieke Wichers.

**Funding acquisition:** Claudia Menne-Lothmann, Jeroen Decoster, Ruud van Winkel, Philippe Delespaul, Marjan Drukker, Marc de Hert, Catherine Derom, Evert Thiery, Bart P. F. Rutten, Nele Jacobs, Marieke Wichers.

**Investigation:** Johanna T. W. Wigman, Sanne H. Booij, Marieke Wichers.

**Methodology:** Anna Kuranova, Johanna T. W. Wigman, Bart P. F. Rutten, Nele Jacobs, Jim van Os, Marieke Wichers.

**Project administration:** Anna Kuranova, Marieke Wichers.

**Resources:** Johanna T. W. Wigman, Claudia Menne-Lothmann, Jeroen Decoster, Ruud van Winkel, Philippe Delespaul, Marjan Drukker, Marc de Hert, Catherine Derom, Bart P. F. Rutten, Nele Jacobs, Jim van Os, Marieke Wichers.

**Software:** Anna Kuranova, Philippe Delespaul, Evert Thiery, Bart P. F. Rutten, Jim van Os.

**Supervision:** Johanna T. W. Wigman, Albertine J. Oldehinkel, Sanne H. Booij, Marieke Wichers.

**Validation:** Johanna T. W. Wigman, Sanne H. Booij, Marieke Wichers.

**Visualization:** Anna Kuranova.

**Writing – original draft:** Anna Kuranova, Johanna T. W. Wigman, Sanne H. Booij, Marieke Wichers.

**Writing – review & editing:** Anna Kuranova, Johanna T. W. Wigman, Claudia Menne-Lothmann, Jeroen Decoster, Ruud van Winkel, Philippe Delespaul, Marjan Drukker, Marc de Hert, Catherine Derom, Evert Thiery, Bart P. F. Rutten, Nele Jacobs, Jim van Os, Albertine J. Oldehinkel, Sanne H. Booij, Marieke Wichers.

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
