## [Decision Letter · Decision Letter 0]

19 Oct 2020

PONE-D-20-11940

Network dynamics of momentary affect states and future course of psychopathology in adolescents

PLOS ONE

Dear Dr. Kuranova,

Thank you for submitting your manuscript to PLOS ONE. After careful consideration, we feel that it has merit but does not fully meet PLOS ONE’s publication criteria as it currently stands. Therefore, we invite you to submit a revised version of the manuscript that addresses the points raised during the review process.

We look forward to receiving your revised manuscript.

Kind regards,

Geilson Lima Santana, M.D., Ph.D.

Academic Editor

PLOS ONE

Journal Requirements:

3. Please ensure that you refer to Figure 0 in your text as, if accepted, production will need this reference to link the reader to the figure.

Reviewers' comments:

Reviewer's Responses to Questions

**Comments to the Author**

1. Is the manuscript technically sound, and do the data support the conclusions?

Reviewer #1: Yes

Reviewer #2: Partly

2. Has the statistical analysis been performed appropriately and rigorously? 

Reviewer #1: Yes

Reviewer #2: No

3. Have the authors made all data underlying the findings in their manuscript fully available?

Reviewer #1: Yes

Reviewer #2: No

4. Is the manuscript presented in an intelligible fashion and written in standard English?

Reviewer #1: Yes

Reviewer #2: Yes

5. Review Comments to the Author

Reviewer #1: Important note: This review pertains only to ‘statistical aspects’ of the study and so ‘clinical aspects’ [like medical importance, relevance of the study, ‘clinical significance and implication(s)’ of the whole study, etc.] are to be evaluated [should be assessed] separately/independently. Further please note that any ‘statistical review’ is generally done under the assumption that (such) study specific methodological [as well as execution] issues are perfectly taken care of by the investigator(s). This review is not an exception to that and so does not cover clinical aspects {however, seldom comments are made only if those issues are intimately / scientifically related & intermingle with ‘statistical aspects’ of the study}. Agreed that ‘statistical methods’ are used as just tools here, however, they are vital part of methodology [and so should be given due importance].

COMMENTS: Your ABSTRACT is well drafted but assay type. Please note that it is preferable [refer to item 1b of CONSORT checklist 2010: Structured summary of trial design, methods, results, and conclusions] to divide the ABSTRACT with small sections like ‘Objective(s)’, ‘Methods’, ‘Results’, ‘Conclusions’, etc. which is an accepted practice of most good/standard journals [including PLOS]. It will definitely be more informative then, I guess [even if your article type is ‘Research Article’].

Please recheck reference 17 [Hartmann JA, Wichers M, van Bemmel AL, Derom C, Thiery E, Jacobs N, et al. The serotonin transporter 5-HTTLPR polymorphism in the association between sleep quality and affect. Eur Neuropsychopharmacol. 2014;24: 1086–1090] because it mainly concludes that “The serotonin transporter 5-HTTLPR polymorphism in the association between sleep quality and affect.” and says the “there was a significant interaction between sleep quality and genotype in predicting positive affect the next day”, therefore, in my opinion, ‘poor sleep quality’ may not be taken as a separate (direct) risk factor as it is likely be covered in ‘genetic risk’ and we are focusing on (or talking about) psychopathology in adolescents [in this age group, biologically reverse association (temporally) is a possibility]. I am not a psychiatrist or psychologist (i.e. subject expert) to pass such a comment, but expressing biological possibility [as this article is nicely presented/written by well-known group].

Account given about sample selection (including regarding sample size) is little confusing. It is said in ‘Methods-Sample and design’ section that “Data were obtained from a longitudinal prospective nested cohort study ‘TWINSSCAN’ (please note that ‘TwinssCan-website’ gives info in other than English language), but further say (immediately) that “a sample recruited from the East-Flanders Prospective Twin Study (EFPTS)” and then say, we selected participants with low quality of childhood experiences. The resulting sample comprised 159 participants. Furthermore, other twins and their (non-twin) siblings between the ages of 15 and 34 could sign up through a general invitation in the twin registry newsletter. The sample size was determined based on previous experience sampling studies”. In ‘Measurements-Quality of childhood experiences’ (n = 451) is quoted [page-7, line-20]. Will you please re-draft this section to make the thigs clearer?

Though I agree with ‘limitations of the study’ brought out in ‘Methodological issues’ (at end of ‘discussion’ section), I guess, epilogue could have been more positive. Figures drawn [including ‘Supplementary Figures’] are nice & very informative [highly appreciable].

Reviewer #2: Title:Network dynamics of momentary affect states and future course of psychopathology in

adolescents

The manuscript by Kuranova and colleagues examines whether adolescents who experience increased psychopathology symptoms over 1 year have different lagged emotion networks than their peers who do not increase in symptoms. There are a lot of strengths of this study including the longitudinal assessment of symptoms, the ‘at-risk’ nature of the population (i.e. adolescents with relatively low childhood happiness). In addition, the authors seem to have thought carefully about how to best construct and interpret the temporal networks. The network analyses yield no statistical differences between the groups, however, they descriptively interpret differences in the networks. Namely, that the ‘Stable’ group’s network does not contain the negative affect cluster and that the lagged edges between negative affect items were stronger in the ‘Increasing’ group, potentially reflecting their engagement in vicious, self-reinforcing cycles. That said, there appeared to be several arbitrary decisions in the methods making inferences (even lack of them) challenging. Specific comments are below

1. Change-score SDs: The standard deviation for the symptom change scores per group is not given. With only the mean, it is difficult for the reader to determine how uniform the groups are. Showing the groups’ T0 and and T1 symptoms does not reflect the within-person change that was the basis of the tertile split. Further, if the authors wish to say the ‘stable group’ contains only those who do not develop any new symptoms, they should demonstrate this with the SD and range of change scores for the Stable group.

a. Relatedly, Figure 1 should have error bars for T0 and T1 symptom group means

2. Group Labels: Throughout the manuscript, the authors refer to the groups as those who develop psychopathology and those who do not develop any symptoms. These “groups” are formed arbitrarily, though. For instance, why did the authors artificially split the SCL90 change? This similarly seems arbitrary and, unless the authors have strong apriori reasons to expect the function to be nonlinear (logistic) at these points, this analysis will artificially reduce plausible associations. Further, did the authors collect data from those with “higher levels of happy childhood experiences” and are not reporting it, or did they just not collect those data? (stratified as an inclusion criteria?). If so this would seem to restrict range artificially. Use of median split is not advisable. Why did the authors do this?

3. EMA item selection: Although the authors provide some justification for only including EMA items that were not highly correlated, it is circular to use an arbitrary correlation cut-off of these EMA items as the basis for including/excluding items from the primary hierarchical regression.

4. Separate HLMS per group: Why were models fitted separately for groups? They should be fitted together with group as a moderator (or even better, change in SCL as a continuous moderator, not excluding any of the data). This would in theory also allow for a statistical test to determine differences in paths by group-status -- the specific question they pose in the introduction of the article

5. Support for PA hypothesis: The authors make an interesting hypothesis about resilience to psychopathology being linked to positive states that affect/interrupt/down-regulate negative states. It would be helpful to mention this earlier in the introduction, perhaps when discussing the network evidence that motivates the ‘vicious cycle’ hypothesis.

6. Adolescence and childhood adversity as interacting risk factors: In paragraph 4, the authors write that it is important to examine emotional dynamics in a pre-clinical, at-risk population, and that adolescents are a well-suited population. In paragraph 5, it is noted that the current sample was adolescents who also have an additional risk factor: low childhood happiness. It seems paragraph 4 could incorporate why adolescents with childhood adversity are a good at-risk population. Other open questions that could be addressed in this paper:

1. It seems like the authors are specifically examining middle and late adolescence, with the mean age being about 17 in both groups. Is this an important distinction?

2. It’s unclear why adults were included in the analyses (the age range was 14-34 years old)

7. Clinical Relevance. Without comparing the groups’ SCL-90 scores to a defined clinical cutoff it is challenging to determine whether an increased score actually reflects the onset of a disorder. If the authors are not suggesting that a clinically significant disorder has manifested, then their language should reflect that they are simply measuring symptom increases. Examples of this language are:

1. “who develop more severe psychopathology over time” (line 11, p5);

2. “individuals who will develop psychopathology over time (Increase group) (line 18, p11);

3. “individuals who will not develop any new symptoms (Stable group)(line 20, p11

8. ESM completion descriptive statistics and group comparison: (1), do participants complete 60 items ten times a day for 6 days? This could be clearer in the methods because that sounds like a lot. (2), it would be helpful if the authors would provide information about avg. number of observations per person (and avg number of consecutive/lagged observations), as well as the range. We know that those with < 30% were excluded but more information could be provided. The number of surveys completed could also be compared across groups as this may influence the networks

9. Transdiagnostic measurement: There is a large movement in the field recognizing the overlap of psychopathology and capturing a ‘p-factor that reflects general risk for/experience of psychopathology broadly. Therefore, it’s not necessarily an issue to analyze the sum score of the SCL-90 and speak about ‘psychopathology’ broadly. However, I think the manuscript may be stronger if the authors explicitly discuss the benefits of examining psychopathology transdiagnositically rather than according to traditional diagnostic categories.

1. It may also be considered as a factor in the lack of statistically significant findings. E.g. increases in certain symptoms may be most strongly linked to the pattern of affective dynamics the authors hypothesized, thus, by looking at all symptom increases equally, that specificity is aggregated over.

10. Racial and Ethnic Diversity: Acknowledgement of the lack of racial and ethnic diversity/generalizability in the sample and any mention of how that could impact the findings would improve the paper.

11. “Resilience”: It seems a little odd to talk about people who maintain symptoms as “resilient”, especially without providing information about the severity of their baseline scores (123, I think) and especially when you have a group that actually decreased in their symptoms over time.

12. Out-strength centrality is an outcome for aim 2 but it is not explicitly defined

13. Formatting, Grammar, Spelling:

1. Methods, p7, line 13: “Sample In addition” is missing a period

2. Methods p9, line 5: “Similar to previous [...], only date from participants…” appears to be missing a word after previous

3. Reference #2 “PDPDDPD”?

4. Reference #22, pg 23, “[doi]” at the end

5. Reference #42, pg 25, double check name and format

6. PLOS authors have the option to publish the peer review history of their article (what does this mean?). If published, this will include your full peer review and any attached files.

Reviewer #1: No

Reviewer #2: No

---

## [Author Response · Author response to Decision Letter 0]

22 Dec 2020

To: Geilson Lima Santana, M.D., Ph.D.

Academic Editor of PLOS ONE

 Groningen, December 2020

Dear Dr. Santana,

We want to thank the Editor and the reviewers for their constructive comments on our paper. We thoroughly revised our manuscript in accordance with the suggestions made, and answered to all thoughts, suggestions, and concerns (see below for a detailed response, organized as point-by-point replies). We believe that this revision has led to substantial improvements of the quality of our article. We hope you will consider our manuscript again, in this enhanced version, for publication in PLOS ONE. 

Kind regards, on behalf of all co-authors,

Anna Kuranova, MD MSc

Comments of the Editor

1. If there are ethical or legal restrictions on sharing a de-identified data set, please explain them in detail (e.g., data contain potentially identifying or sensitive patient information) and who has imposed them (e.g., an ethics committee). Please also provide contact information for a data access committee, ethics committee, or other institutional body to which data requests may be sent. We will update your Data Availability statement on your behalf to reflect the information you provide.

Reply: As there is a possibility to identify participants based on their clinical and experience sampling data, the datasets generated and/or analyzed during the current study cannot be made publicly available based on European law. The study was approved by the local ethics committee (KU Leuven, Nr. B32220107766), which has also imposed the data availability restrictions. Data requests may be sent to the TWINSSCAN general contact email address "info@twinsscan.eu”. This information can be included in the paper, if required. 

2. Please ensure that you refer to Figure 0 in your text as, if accepted, production will need this reference to link the reader to the figure.

Reply: The reference to Figure 0 is added to the text (see page 10).

Reply: The captions are now included and the citations have been updated accordingly. 

Reviewer #1. 

1. Important note: This review pertains only to ‘statistical aspects’ of the study and so ‘clinical aspects’ [like medical importance, relevance of the study, ‘clinical significance and implication(s)’ of the whole study, etc.] are to be evaluated [should be assessed] separately/independently. Further please note that any ‘statistical review’ is generally done under the assumption that (such) study specific methodological [as well as execution] issues are perfectly taken care of by the investigator(s). This review is not an exception to that and so does not cover clinical aspects {however, seldom comments are made only if those issues are intimately / scientifically related & intermingle with ‘statistical aspects’ of the study}. Agreed that ‘statistical methods’ are used as just tools here, however, they are vital part of methodology [and so should be given due importance].

Your ABSTRACT is well drafted but assay type. Please note that it is preferable [refer to item 1b of CONSORT checklist 2010: Structured summary of trial design, methods, results, and conclusions] to divide the ABSTRACT with small sections like ‘Objective(s)’, ‘Methods’, ‘Results’, ‘Conclusions’, etc. which is an accepted practice of most good/standard journals [including PLOS]. It will definitely be more informative then, I guess [even if your article type is ‘Research Article’].

Reply: Thank you for this suggestion. The abstract has been rewritten based on item 1b of the CONSORT checklist 2010, inasmuch as applicable (the CONSORT checklist describes trial designs and this is an observational study), please see below: 

Background: Recent theories argue that an interplay between (i.e., network of) experiences, thoughts and affect in daily life may underlie the development of psychopathology.

Objective: To prospectively examine whether network dynamics of everyday affect states are associated with a future course of psychopathology in adolescents at an increased risk of mental disorders.

Methods: 159 adolescents from the East-Flanders Prospective Twin Study cohort participated in the study. At baseline, their momentary affect states were assessed using the Experience Sampling Method (ESM). The course of psychopathology was operationalized as the change in the Symptom Checklist-90 sum score after 1 year. Two groups were defined: one with a stable level (n=81) and one with an increasing level (n=78) of SCL-symptom severity. Group-level network dynamics of momentary positive and negative affect states were compared between groups.

Results: The group with increasing symptoms showed a stronger connections between negative affect states and their higher influence on positive states, as well as higher proneness to form ‘vicious cycles’, compared to the stable group. Based on permutation tests, these differences were not statistically significant.

Conclusion: Although not statistically significant, some qualitative differences were observed between the networks of the two groups. More studies are needed to determine the value of momentary affect networks for predicting the course of psychopathology. 

2. Please recheck reference 17 [Hartmann JA, Wichers M, van Bemmel AL, Derom C, Thiery E, Jacobs N, et al. The serotonin transporter 5-HTTLPR polymorphism in the association between sleep quality and affect. Eur Neuropsychopharmacol. 2014;24: 1086–1090] because it mainly concludes that “The serotonin transporter 5-HTTLPR polymorphism in the association between sleep quality and affect.” and says the “there was a significant interaction between sleep quality and genotype in predicting positive affect the next day”, therefore, in my opinion, ‘poor sleep quality’ may not be taken as a separate (direct) risk factor as it is likely be covered in ‘genetic risk’ and we are focusing on (or talking about) psychopathology in adolescents [in this age group, biologically reverse association (temporally) is a possibility]. I am not a psychiatrist or psychologist (i.e. subject expert) to pass such a comment, but expressing biological possibility [as this article is nicely presented/written by well-known group].

Reply: thank you for pointing out this issue. After reconsidering it, we agree that the article by Hartmann and colleagues is indeed not the optimal reference for the association between sleep quality and altered affect. Although the main effect of the sleep quality on next day affect was present in the model, the results primarily concern the interaction effect of the serotonin transporter 5-HTTLPR polymorphism and the sleep quality and not the main effect per se. Therefore, we deleted this reference here, and now cite the review by Tempesta and colleagues [1] as well as articles by Blaxton and colleagues [2] and Sin and colleagues [3] to support the claim about the sleep quality being associated with altered affect. 

3. Account given about sample selection (including regarding sample size) is little confusing. It is said in ‘Methods-Sample and design’ section that “Data were obtained from a longitudinal prospective nested cohort study ‘TWINSSCAN’ (please note that ‘TwinssCan-website’ gives info in other than English language), but further say (immediately) that “a sample recruited from the East-Flanders Prospective Twin Study (EFPTS)” and then say, we selected participants with low quality of childhood experiences. The resulting sample comprised 159 participants. Furthermore, other twins and their (non-twin) siblings between the ages of 15 and 34 could sign up through a general invitation in the twin registry newsletter. The sample size was determined based on previous experience sampling studies”. In ‘Measurements-Quality of childhood experiences’ (n = 451) is quoted [page-7, line-20]. Will you please re-draft this section to make the thigs clearer?

Reply: We understand the confusion and have adapted the section about sample selection to make the procedure and sample size more clearly. Please see below: 

Methods, Page 6: “Data were obtained from the longitudinal prospective study ‘TWINSSCAN’ (“http://www.twinsscan.eu”; website only in Dutch), a cohort nested in the East-Flanders Prospective Twin Study (EFPTS), a register of all multiple births in the Province of East Flanders, Belgium, from 1964 onwards [4,5]. In 2010 potential participants for the TWINSSCAN cohort were recruited by sending invitation letters to all EFPTS participants who were between the ages of 15 and 18 years. Furthermore, to recruit more twins and their non-twin siblings between ages of 15 and 34, a general invitation was included in a newsletter from the EFPTS. All participants provided their written informed consent. For those participants who were aged below 18 years, their parents or caretakers provided additional written consent. The local ethics committee (KU Leuven, Nr. B32220107766) approved the study. 

The TWINSSCAN sample enrolled in the baseline assessment comprised 839 people and involved a broad range of measurements, including clinical interviews, questionnaires, experiments and an ESM period [6]. For the current paper, additional inclusion criteria applied based on available data. First, participants needed to score below the median on items assessing the happiness of their childhood (see Measures for more details), leading to the exclusion of 388 individuals. Second, they needed complete data on the Symptom Check List-90 (SCL-90) [7] at both baseline (T0) and a follow-up wave after one year (T1), leading to the exclusion of another 202 individuals. Third, we excluded 10 individuals with more than 30% missing ESM data points. Altogether, this resulted in a sample of 239 participants, who were grouped according to their pattern of SCL-90 symptom change over one year (see details below). This change score was divided into tertiles, representing groups with decreasing, stable and increasing levels of symptoms. The group with decreasing symptoms was excluded for theoretical reasons (see details below), resulting in a final sample of 159 individuals, categorized into a group with Stable symptom levels (n=81) and a group with Increasing symptom levels (n=78).” 

Methods section, JTV description, page 7: “…a median split of the sum score of the four items was used to define a high-scoring and low-scoring individuals on safe and happy childhood experiences, of whom the former were excluded from further analysis (see ‘Sample and design’).”

4. Though I agree with ‘limitations of the study’ brought out in ‘Methodological issues’ (at end of ‘discussion’ section), I guess, epilogue could have been more positive. Figures drawn [including ‘Supplementary Figures’] are nice & very informative [highly appreciable].

Reply: thank you for appreciation of our work. We have changed the final paragraph of the paper to give a more positive perspective on our findings and future directions. 

Discussion section, page 21: “Finally, this study was conducted at the group level, as we examined average differences in the structures of group dynamic networks, and therefore it is not possible to directly estimate the effects for individual cases. This would be an important next step. Identification of dynamical patterns at the individual level and connecting those to future changes in symptoms will add new evidence on the relevance of the network approach to affect dynamics and may yield promising targets for future personalized diagnostic, prevention and treatment strategies [8–10].

Taken together, our findings cautiously suggest that some differences in dynamical networks of affect states of adolescents with different mental health trajectories may exist already one year before new symptoms develop. However, these differences may be subtle and not yet statistically detectable by the permutation testing approach. Hence, more studies examining these qualitative indicators at an early stage are needed to give a more definite answer as to whether these emotion dynamics can be detected in a reliable way, and if so, how they may be used to create new methods of treatment and prevention of psychopathology.”

Reviewer #2. 

The manuscript by Kuranova and colleagues examines whether adolescents who experience increased psychopathology symptoms over 1 year have different lagged emotion networks than their peers who do not increase in symptoms. There are a lot of strengths of this study including the longitudinal assessment of symptoms, the ‘at-risk’ nature of the population (i.e. adolescents with relatively low childhood happiness). In addition, the authors seem to have thought carefully about how to best construct and interpret the temporal networks. The network analyses yield no statistical differences between the groups; however, they descriptively interpret differences in the networks. Namely, that the ‘Stable’ group’s network does not contain the negative affect cluster and that the lagged edges between negative affect items were stronger in the ‘Increasing’ group, potentially reflecting their engagement in vicious, self-reinforcing cycles. That said, there appeared to be several arbitrary decisions in the methods making inferences (even lack of them) challenging. Specific comments are below.

Reply: thank you for your appreciation of our work and your feedback.

1. Change-score SDs: The standard deviation for the symptom change scores per group is not given. With only the mean, it is difficult for the reader to determine how uniform the groups are. Showing the groups’ T0 and T1 symptoms does not reflect the within-person change that was the basis of the tertile split. Further, if the authors wish to say the ‘stable group’ contains only those who do not develop any new symptoms, they should demonstrate this with the SD and range of change scores for the Stable group.

a. Relatedly, Figure 1 should have error bars for T0 and T1 symptom group means

Reply: We agree that this additional information is valuable for the reader to see, and extended the information presented in Table 1 with the means, SDs and ranges of the SCL-90 change scores per group (please see below). Concerning the “stable group”, we would like to stress that we do not mean that none of participants belonging to this group developed any symptoms. The group was labeled “Stable” because in this dataset the second tertile had the smallest change in the level of symptoms. It should be noted, however, that for the research questions not the exact change of the symptoms for each group is relevant but the fact that groups of interest had the similar level of symptoms at baseline and different at follow-up, with one group (labeled as “Increase”) had a significantly higher level of symptoms. This way we were able to test whether the group networks differed before the differences in the level of symptoms emerged. Now we adapted the description of the groups and added more information, please see below. Concerning Figure 1, SDs are now have been added to a graph, please see in the separate file.

Group composition, Methods, page 9: “To assess change in the level of symptoms, we subtracted the SCL-90 scores at T0 from the SCL-90 scores at T1 for each participant. After that, these change scores were divided into tertiles, resulting in 3 groups defined by a reduction (Decrease group, mean SCL-90 sum score change = -41.48 points, SD =33.09, n = 80;), minimal change (Stable group, mean SCL-90 sum score change = -5.02 points, SD = 4.95, n = 81) and an increase in symptom level (Increase group, mean SCL-90 sum score change = 25.66, SD = 22.5, n = 78).”

2. Group Labels: Throughout the manuscript, the authors refer to the groups as those who develop psychopathology and those who do not develop any symptoms. These “groups” are formed arbitrarily, though. For instance, why did the authors artificially split the SCL90 change? This similarly seems arbitrary and, unless the authors have strong apriori reasons to expect the function to be nonlinear (logistic) at these points, this analysis will artificially reduce plausible associations. Further, did the authors collect data from those with “higher levels of happy childhood experiences” and are not reporting it, or did they just not collect those data? (Stratified as an inclusion criterion?). If so this would seem to restrict range artificially. Use of median split is not advisable. Why did the authors do this?

Reply: First, the subgroups were created for practical reasons: we were interested in the predictive validity of dynamic affect state networks and in the possibility of visual assessment and comparison of such networks (e.g. to assess the presence of “vicious cycles”). For that, either individual or group networks can be created based on ESM data. However, creation of individual networks of 6 affect states requires a larger number of observations per person [11] which was not possible with current dataset. Therefore we chose to construct group networks, in line with previous studies [12–15] as well as in our previously published study using the same dataset [16]. 

To construct the group networks, both the number of observations per individual and the group size are important. The largest possible group size (and simultaneously trying to keep similar size for the subgroups that will be compared) can be achieved by splitting the data in equal groups. However, given the theoretical reasoning for not using a group with higher level of symptoms at baseline (we were interested in the prediction of more severe (not less severe) levels of psychopathology), we chose to use tertile split and to exclude those with decreasing levels of psychopathology. We then tested whether the resulting two subgroups differed in their levels of symptoms at baseline and follow-up to confirm that these two groups have the same level of symptoms at baseline and different level of symptoms at follow-up. 

However, we agree that the tertile split is an arbitrary decision and that other group compositions are indeed possible. Therefore, to investigate whether the visual differences found in the third research question were robust, we now checked the effect of alternative grouping and performed a restricted multiverse analysis to explore this issue (based on the idea by Steegen, Tuerlinckx and collegues 2016 [17]). For that, we analyzed all possible combinations of cutoffs of SCL-90 change scores, with the following conditions: each group should have (i) at least 70 people (power restriction), (ii) comparable levels of SCL-90 scores and mean level of all six affect states at baseline, and (iii) different levels of SCL-90 scores at follow-up. This approach led to 29 possible combinations of groupings. We created 29 pairs of networks for 29 combinations of groups (please see supplementary document 3 for all networks) and visually inspected these. From these 29 networks, the networks for an “increase group” did not differ between each other. The networks for “Stable” groups had more variations but (almost) all had the similar structure to the one reported in the main analysis and fitted the pattern of (almost) absence of “vicious” cycles, fewer negative clusters and connections and more downregulating connections from positive cluster to negative nodes. Specifically, among the networks of the “stable” groups, only one (~3.5%) contained the possibility for a “vicious cycle”; eight (~27.6%) upregulating connections between any 3 negative nodes (without forming self-reinforcing loops, i.e. two connections and three nodes, e.g. from “Lonely” to “Down” and from “Irritated” to “Down”); 14 (~48.3%) upregulating connections between any 2 negative nodes (i.e. one connection between two nodes, e.g. from “Lonely” to “Down”; with the exception of the network with “vicious cycle” containing two connections between two negative nodes); and seven (~24.1%) no connections between negative nodes and therefore no negative cluster at all. Moreover, all networks of both groups contained downregulating connections from a positive cluster to at least one negative node but of the 29 networks of the “stable” groups, 11 (~37.9%) contained two downregulating connections from positive cluster to negative nodes, whereas all the networks of “increase” groups contained only one such connection. 

Thus, despite different cut-offs, resulting networks of “Increase” and “Stable” groups showed the similar compositions and dynamic patterns to the ones reported in the main findings. Therefore, we argue that, although our results are based on a cut-off that is somewhat arbitrary, they are robust against changes in this cut-off. We have added a short description of the restricted multiverse analysis to the Methods section and Supplementary materials 3. We also incorporated this addition in the results and discussion of our paper, please see below. 

Methods, page 13: “…to ensure the robustness of the results of the visual inspection, we performed a limited version of multiverse analysis (based on [17]) to test the influence of different group compositions based on different cut-offs for the SCL-90 change score. A detailed explanation of the calculations, the visualization, the assessment and the limited multiverse analysis can be found in the supplementary materials (S3 text, S4 text, S5 Table, S7 Figure).”

Results, page 16: “The negative nodes in the Stable group, however, were not connected and could therefore not form a vicious cycle. These networks differences were robust to the changes in group allocations based on the limited multiverse analysis (see S4 text and S5 Table for details).”

Discussion, page 20: “…The limited multiverse analysis that we ran to investigate the potential effect of our subgroup selection strengthens us in our assumption that our choice was solid (see S4 text and S5 Table).”

Second, concerning the median split for “happy childhood experiences”, we followed a similar theoretical reasoning. We sought the optimal balance between maintaining a high number of people in the selection and at the same time a relatively low level of happy childhood experiences. Therefore, we included only the 50% with lower levels of JTV scores. However, we agree that this decision is indeed somewhat arbitrary and acknowledge this now in the Methodological issues. Moreover, now we changed the description of the sample in the Methods section, so the procedure is clearer and mention the issue in the Discussion, please see below: 

Methods, page 7: “Altogether, this resulted in a sample of 239 participants, who were grouped according to their pattern of SCL-90 symptom change over one year (see details below). This change score was divided into tertiles, representing groups with decreasing, stable and increasing levels of symptoms. The group with decreasing symptoms was excluded for theoretical reasons (see details below), resulting in a final sample of 159 individuals, categorized into a group with Stable symptom levels (n=81) and a group with Increasing symptom levels (n=78).” 

Discussion, page 19-20: “Second, we made several methodological decisions that may have impacted the results. (I) The sample was created by selecting the 50% of people with the lowest level of happy childhood experiences, and the SCL-90 change scores were split into tertiles. Although these decisions are, to a certain extent, arbitrary, they were based on theoretical (e.g., interest in those at highest risk) as well as methodological (e.g., optimising subgroup size) reasons. In addition, the results were robust to changes in group allocations, supporting our confidence in the choices made”. 

3. EMA item selection: Although the authors provide some justification for only including EMA items that were not highly correlated, it is circular to use an arbitrary correlation cut-off of these EMA items as the basis for including/excluding items from the primary hierarchical regression.

Reply: We had several reasons for choosing the items that we did; we would like to take this opportunity to explain our reasoning more clearly.

 First, in order to create networks, we used VAR models, i.e. a set of six model equations. If we had chosen the items based on which items were included or excluded in the primary hierarchical regression analysis instead of based on correlations, we would have had to do this for all six models. This would probably have led to different items compositions for each model, which would render creation of networks impossible. 

Second, we chose EMA items based on several criteria, of which low inter-item correlations was only one. As we were interested in affect networks, we selected at least one item from each quadrant between the axes of “pleasure” and “arousal” as defined in the circumplex model of affect [18,19]. The items “Down” and Energetic” were added due to their theoretical relevance. Additionally, we aimed to avoid the problem of floor effects in case of items with insufficient variation over time. Finally, we selected items with comparable means and variances (no more than 10% difference for within-person SDs) to ensure that the resulting differences in networks are not due to different item means and variances. After all these criteria were met, the only choice left was between the two items “relaxed” and “satisfied”, and this is where the correlation criterion was applied. 

We now explain our rationale about the item selection more thoroughly in the Methods section: 

Methods, page 9-10: “We selected ESM items based on both theoretical and methodological criteria. First, we only selected experiential affect states (not thoughts, behaviors or context information). Second, of these, we selected at least one item from each quadrant between the axes of “pleasure” and “arousal” as defined in the circumplex model of affect [18,19]. Additionally, we added the items “Down”, and an item “Energetic”, as they reflect common transdiagnostic symptoms [20,21]. Third, to avoid a floor effect because of insufficient variance [22], we chose items with a within-person standard deviation (SD) of around 1.0 (see Table 1). Fourth, we chose affect states that were not highly correlated with each other (r < 0.5), so that all items captured different aspects of a momentary mental experience. Fifth, to ensure that the differences between group networks originated from differences in the dynamics between affect states, we checked whether the mean levels of the selected items did not differ between the Increase and Stable groups, and whether the within-person SDs of the selected items did not differ more than 10-12%.”

4. Separate HLMS per group: Why were models fitted separately for groups? They should be fitted together with group as a moderator (or even better, change in SCL as a continuous moderator, not excluding any of the data). This would in theory also allow for a statistical test to determine differences in paths by group-status -- the specific question they pose in the introduction of the article

Reply: This is an interesting suggestion. Fitting the models with group as a moderator would indeed allow for determining the effect of subgroup (or continuous SCL change scores) on associations between each affect state at t-1 and itself and others at t in each model. However, our research question was aimed at investigating the predictive validity of network analysis, and specifically of specific networks characteristics (connectivity, centrality, the effect of one cluster on another) that are thought to reflect general properties of networks as a whole. Beta-coefficients from the models represent individual edges of the networks; adding group status or level of symptoms as moderator will be possible only on the level of these individual edges, but not on the level of these composite network characteristics. For example, the negative connectivity characteristic is a sum of absolute values of six different edges, and it is possible that for some of these edges the moderation effect is opposite. Therefore, to answer our research question, we needed to construct networks first and then assess differences between the characteristics of each network with permutation tests. Therefore, although the use of the group as moderator in each model is interesting, this approach answers a different research question.

We now make the aim of specifically the network comparison clearer throughout the manuscript: 

Introduction, page 5: “We hypothesize that affect state networks of individuals who are vulnerable to the development of future psychopathology will show dynamics of affect states that are prone to the development of vicious cycles. For such individuals, negative affect states will have strong mutually reinforcing connections. Furthermore, we hypothesize that in networks of individuals who are resilient against psychopathology (i.e. do not develop new or more severe symptoms despite being at an increased risk), positive affect states have the potential to interfere with such vicious cycles by down-regulating one or more of these negative affect states.”

Discussion, page 16: “The purpose of this study was to investigate whether the presence of differences in the dynamic networks of momentary affect states precedes the development of more severe psychopathological symptoms in adolescents at an increased risk.”

5. Support for PA hypothesis: The authors make an interesting hypothesis about resilience to psychopathology being linked to positive states that affect/interrupt/down-regulate negative states. It would be helpful to mention this earlier in the introduction, perhaps when discussing the network evidence that motivates the ‘vicious cycle’ hypothesis.

Reply: Thank you for this suggestion. The part about the “vicious cycle” now reads:

Introduction, page 4: “For example, for some people, feeling lonely may induce states of feeling down and irritated. These affect states, in turn, may re-activate feeling lonely. Such mutual influences, when occurring repeatedly, can lead to ‘vicious cycles’ of affect states that keep reinforcing each other, trapping a person in a negative flow. Yet, for others, feeling lonely may pass without activating other negative affect states, or may be neutralized by a later positive affect state (e.g. feeling cheerful after seeking for social support from peers). Moreover, the ability of positive states to interrupt or downregulate the negative “vicious cycles” may be associated with resilience to psychopathology and may represent an important part of its mechanism. Thus, the impact of a minor mood perturbation may vary depending on the dynamics of affect states. To investigate these dynamics, we need to assess the whole system of interacting positive and negative affect states.” 

6. Adolescence and childhood adversity as interacting risk factors: In paragraph 4, the authors write that it is important to examine emotional dynamics in a pre-clinical, at-risk population and that adolescents are a well-suited population. In paragraph 5, it is noted that the current sample was adolescents who also have an additional risk factor: low childhood happiness. It seems paragraph 4 could incorporate why adolescents with childhood adversity are a good at-risk population. 

Reply: Thank you for this suggestion. Paragraph 4 now reads: 

Introduction, page 5: “To determine whether characteristics of the dynamics between momentary affect states are key factors in the developmental process of symptom formation, we need to examine whether these characteristics are already present in populations at increased risk for psychopathology, before more severe symptoms arise. The reasoning behind including individuals at increased risk is that any underlying vulnerability for, as well as resilience against, psychopathology can be exposed only when challenged by risk factors. Because (i) adolescence is a sensitive period for the development of psychopathology in which symptoms often emerge for the first time [23,24], and (ii) a low level of happy childhood experiences is a known risk factor for psychopathology [25,26], adolescents with low levels of happy childhood experiences represent a well-suited population for this purpose. 

Therefore, we aim in this paper to explore whether the dynamic network structure of affect states differs between adolescents who develop more severe psychopathology over time and adolescents who do not develop any new symptoms. We used a prospective research design in an adolescent population with experience sample (ESM) data collection carried out at baseline and with follow-up assessments to differentiate the course of future psychopathology.”

6.5. Other open questions that could be addressed in this paper:

1. It seems like the authors are specifically examining middle and late adolescence, with the mean age being about 17 in both groups. Is this an important distinction?

Reply: No, this mean age is only due to chance. During the recruitment procedure all participants of the Twin Registry from 15 to 34 years old received letters of recruitment or an invitation in the newsletter.

2. It’s unclear why adults were included in the analyses (the age range was 14-34 years old)

Reply: Invitations for the TWINSSCAN study were sent out to all twins of the Twin Registry from 15 to 34 years (and to their other siblings and parents). Although the large majority of our participants were younger, several emerging adults were also included in the TWINSSCAN cohort. We decided to keep their data in the analysis because we were interested in the sensitive period for the development for psychopathology, which is broader than adolescence [23,27]. Moreover, keeping these participants slightly increased the power and the consistency with other studies using the same dataset. Because the mean age of our subsample was 17.46 years, we have kept the word ‘adolescents’ throughout the manuscript. We have added these considerations to the discussion of the study limitations, please see below: 

Discussion, page 19: “(…) (III) although most participants were adolescents (mean age = 17.46), emerging adults were also included in the TWINSSCAN cohort. We decided to keep their data in the analysis because we were interested in the sensitive period for the development for psychopathology, which is broader than adolescence per se [23,27]. Moreover, keeping these participants slightly increased the power of our study and its consistency with other studies using the same dataset.”

7. Clinical Relevance. Without comparing the groups’ SCL-90 scores to a defined clinical cutoff it is challenging to determine whether an increased score actually reflects the onset of a disorder. If the authors are not suggesting that a clinically significant disorder has manifested, then their language should reflect that they are simply measuring symptom increases. Examples of this language are:

1. “who develop more severe psychopathology over time” (line 11, p5);

2. “individuals who will develop psychopathology over time (Increase group) (line 18, p11);

3. “individuals who will not develop any new symptoms (Stable group)(line 20, p11)

Reply: Thank you for pointing out this nuance. Indeed, although the difference in the level of symptoms between the groups at follow-up roughly corresponded to one severity category [28], we do not claim the onset of any disorders and just describe the change in the level of symptoms. We have adapted the wording to make this clearer, please see examples below: 

Introduction, page 5: “who develop higher level of symptoms over time”

Introduction, page 6: “Stated specifically in terms of network characteristics, we expect that the network of affect states in adolescents with a future increase in the level of symptoms compared to the network of affect states of adolescents with a relatively stable symptom level”

Methods, page 12: “individuals who will develop more symptoms over time (Increase group) (…) Individuals with the relatively stable level of symptoms (Stable group)”

Discussion, page 16: “In this study, we examined, both statistically and descriptively, whether differences in the dynamical networks of affect states at baseline can be found between groups of adolescents with increasing and relatively stable levels of psychopathological symptoms over one year.”

8. ESM completion descriptive statistics and group comparison: (1), do participants complete 60 items ten times a day for 6 days? This could be clearer in the methods because that sounds like a lot. (2), it would be helpful if the authors would provide information about avg. number of observations per person (and avg number of consecutive/lagged observations), as well as the range. We know that those with < 30% were excluded but more information could be provided. The number of surveys completed could also be compared across groups as this may influence the networks

Reply: 1) Thank you for pointing out this issue. Participants filled in different amount of items in mornings, evenings and during the day. On average they filled 40 items. We changed the description in the Methods, please see below. 2) The information on the average number of observations and of two consecutive non-missing observations per person is added to Table 1; these parameters did not differ between groups. Concerning the number of completed surveys, it is identical to the number of non-missing observations because we only included observations with all ESM items present in the analysis. Now we added this information to the Methods sections: 

Methods, page 9: “For six days, participants completed short questionnaires (around 40 items, with additional items on mornings and evenings) about their current affect states, thoughts, daily life context and behavior. (… ) Only observations with all present ESM items were included in the analysis.”

9. Transdiagnostic measurement: There is a large movement in the field recognizing the overlap of psychopathology and capturing a ‘p-factor that reflects general risk for/experience of psychopathology broadly. Therefore, it’s not necessarily an issue to analyze the sum score of the SCL-90 and speak about ‘psychopathology’ broadly. However, I think the manuscript may be stronger if the authors explicitly discuss the benefits of examining psychopathology transdiagnositically rather than according to traditional diagnostic categories.

It may also be considered as a factor in the lack of statistically significant findings. E.g. increases in certain symptoms may be most strongly linked to the pattern of affective dynamics the authors hypothesized, thus, by looking at all symptom increases equally, that specificity is aggregated over.

Reply: Thank you for sharing this idea. Indeed, it is possible that looking at psychopathology broadly may have influenced the probability of finding some patterns of affect dynamics specific for certain symptoms. Now we add this consideration as well as further explanations of why we examinedpsychopathology transdiagnostically to the Discussion section. 

Discussion, page 20: “(III) we used the total score of the SCL-90 as an indicator of general psychopathological severity; this could also have led to averaging out any changes in specific areas (e.g., depression). However, using a general index is in line with current views in the field of psychopathology as a broad, transdiagnostic or even one general factor [29,30]. In addition, because this sample is at risk but not diagnosed for any particular psychiatric disorder), we feel that the use of a general index is most suitable.”

10. Racial and Ethnic Diversity: Acknowledgement of the lack of racial and ethnic diversity/generalizability in the sample and any mention of how that could impact the findings would improve the paper.

Reply: We agree with the reviewer and have added an acknowledgment of the lack of ethnic diversity to the “Methodological issues” section: 

Discussion, page 19: “(II) the sample consisted almost exclusively of Caucasian participants, which limits the generalizability of the finding to other populations.”

11. “Resilience”: It seems a little odd to talk about people who maintain symptoms as “resilient”, especially without providing information about the severity of their baseline scores (123, I think) and especially when you have a group that actually decreased in their symptoms over time.

Reply: in this study we understand resilience as maintaining good mental health despite being at increased risk for developing psychopathology. Because the Decrease group had significantly higher starting levels of symptoms than two other groups (M = 168.3), and considering that their level of symptoms roughly corresponded to the “high” category in the SCL-90 manual [28], they do not fit this definition of being resilient. Regarding the reviewer’s consideration about the baseline score, we agree and provide now more information about their severity in Results section, please see below:

Results, Groups, page 13: “The final sample (n=239) was grouped based on tertiles of change in their psychopathological trajectory over the course of one year. This led to three groups: a Stable group (n = 81) with a relatively small decrease in symptoms (for details see Table 1);, an Increase group (n = 78) with a relatively large increase in symptoms (for details see Table 1), and a Decrease group (n = 80), with a relatively large decrease in symptoms (Mage=17.84, age range: 14-33 years, SD = 3.84; 66.25% females). As the latter subgroup had significantly (p<.0001) higher SCL-90 scores at baseline (mean level 168.3, corresponding to “high” symptom level in the normal population [28]) than the other two groups, ,i this group was excluded from analyses. The Stable and the Increase group did not differ significantly on the baseline SCL-90 score (mean level of SCL-90 for the Stable group = 126.8, for the Increase group = 130.24, difference = 3.44, p = .48), and their levels correspond to “mean”/”above mean” levels in a normal population [28]. At T1, the level of symptoms of the Increase group was equal 155.90 (corresponding to “high” levels in the normal population [28]) and significantly higher than that of the Stable group (mean level 121.78) with difference = 34.13, p<0.001 which roughly corresponds to an increase of one severity category [28]. Trajectories of psychopathology for the two groups are presented in Figure 1.” 

Moreover, now we add the explanation about the understanding of resilience to the Introduction section, please see below: 

Introduction, page 5: “Furthermore, we hypothesize that in networks of individuals who are resilient against psychopathology (i.e. do not develop the new or more severe symptoms despite being at an increased risk), positive affect states …”

12. Out-strength centrality is an outcome for aim 2 but it is not explicitly defined

Reply: Thank you for noticing this. We now also define “out-strength centrality” in the Methods section:

Methods, page 12: “(i) the relative importance of the positive nodes in the networks, based on their out-strength centrality measures; (ii) the overall effect of the positive states (‘cheerful’, ‘relaxed’, ‘energetic’) on the negative states (‘irritated’, ‘down’, ‘lonely’) and vice-versa. Out-strength centrality measure is a network characteristic that equals the sum of all connections going from the node of interest to the other nodes, and reflects the overall influence of this node on the other ones. Specifically, the out-strength centrality was calculated by summing the b-coefficients from the regression models for the indicated paths.”

13. Formatting, Grammar, Spelling:

1. Methods, p7, line 13: “Sample In addition” is missing a period

2. Methods p9, line 5: “Similar to previous [...], only date from participants…” appears to be missing a word after previous

3. Reference #2 “PDPDDPD”?

4. Reference #22, pg 23, “[doi]” at the end

5. Reference #42, pg 25, double check name and format

Reply: thank you for noticing these issues. They are corrected in the revised manuscript. 

References:

1. Tempesta D, Socci V, De Gennaro L, Ferrara M. Sleep and emotional processing. Sleep Medicine Reviews. 2018. doi:10.1016/j.smrv.2017.12.005

2. Blaxton JM, Bergeman CS, Whitehead BR, Braun ME, Payne JD. Relationships among nightly sleep quality, daily stress, and daily affect. Journals Gerontol - Ser B Psychol Sci Soc Sci. 2017. doi:10.1093/geronb/gbv060

3. Sin NL, Almeida DM, Crain TL, Kossek EE, Berkman LF, Buxton OM. Bidirectional, Temporal Associations of Sleep with Positive Events, Affect, and Stressors in Daily Life Across a Week. Ann Behav Med. 2017. doi:10.1007/s12160-016-9864-y

4. Loos R, Derom C, Vlietinck R, Derom R. The East Flanders Prospective Twin Survey (Belgium): A population-based register. Twin Res. 1998;1: 167–175. doi:10.1375/136905298320566131

5. Loos R, Derom C, Vlietinck R, Derom R. The East Flanders Prospective Twin Survey (Belgium): A population-based register. Twin Res. 1998;1: 167–175. doi:10.1375/136905298320566131

6. Pries LK, Guloksuz S, Menne-Lothmann C, Decoster J, Winkel R van, Collip D, et al. White noise speech illusion and psychosis expression: An experimental investigation of psychosis liability. PLoS One. 2017;12. doi:10.1371/journal.pone.0183695

7. Derogatis L. SCL-90. Administration, scoring & procedures manual-I for the (revised) version and other instruments of the psychopathology rating scale series. Balt MD Clin Psychom Res Unit, Johns Hopkins Univ Sch Med. 1977. 

8. Bos FM, Snippe E, Bruggeman R, Wichers M, van der Krieke L. Insights of patients and clinicians on the promise of the experience sampling method for psychiatric care. Psychiatr Serv. 2019. doi:10.1176/appi.ps.201900050

9. Kramer I, Simons CJP, Hartmann JA, Menne-Lothmann C, Viechtbauer W, Peeters F, et al. A therapeutic application of the experience sampling method in the treatment of depression: A randomized controlled trial. World Psychiatry. 2014. doi:10.1002/wps.20090

10. von Klipstein L, Riese H, van der Veen DC, Servaas MN, Schoevers RA. Using person-specific networks in psychotherapy: challenges, limitations, and how we could use them anyway. BMC Med. 2020. doi:10.1186/s12916-020-01818-0

11. Rosmalen JGM, Wenting AMG, Roest AM, de Jonge P, Bos EH. Revealing Causal Heterogeneity Using Time Series Analysis of Ambulatory Assessments. Psychosom Med. 2012;74: 377–386. doi:10.1097/PSY.0b013e3182545d47

12. Kelleher I, Wigman JTW, Harley M, O’Hanlon E, Coughlan H, Rawdon C, et al. Psychotic experiences in the population: Association with functioning and mental distress. Schizophr Res. 2015;165: 9–14. doi:10.1016/j.schres.2015.03.020

13. Hasmi L, Drukker M, Guloksuz S, Menne-Lothmann C, Decoster J, van Winkel R, et al. Network approach to understanding emotion dynamics in relation to childhood trauma and genetic liability to psychopathology: Replication of a prospective experience sampling analysis. Front Psychol. 2017;8. doi:10.3389/fpsyg.2017.01908

14. Isvoranu A-M, van Borkulo CD, Boyette L-L, Wigman, Vinkers CH, Borsboom D, et al. A network approach to psychosis: Pathways between childhood trauma and psychotic symptoms. Schizophrenia bulletin. 2016. p. sbw055. doi:10.1093/schbul/sbw055

15. Wigman JTW, De Vos S, Wichers M, Van Os J, Bartels-Velthuis AA. A transdiagnostic network approach to psychosis. Schizophr Bull. 2017. doi:10.1093/schbul/sbw095

16. Kuranova A, Booij SH, Menne-Lothmann C, Decoster J, Van Winkel R, Delespaul P, et al. Measuring resilience prospectively as the speed of affect recovery in daily life: A complex systems perspective on mental health. BMC Med. 2020. doi:10.1186/s12916-020-1500-9

17. Steegen S, Tuerlinckx F, Gelman A, Vanpaemel W. Increasing Transparency Through a Multiverse Analysis. Perspect Psychol Sci. 2016. doi:10.1177/1745691616658637

18. Feldman Barrett L, Russell JA. Independence and bipolarity in the structure of current affect. J Pers Soc Psychol. 1998;74: 967–984. doi:10.1037/0022-3514.74.4.967

19. Russell JA, Weiss A, Mendelsohn GA. Affect Grid: A Single-Item Scale of Pleasure and Arousal. J Pers Soc Psychol. 1989;57: 493–502. doi:10.1037/0022-3514.57.3.493

20. Harvey SB, Wessely S, Kuh D, Hotopf M. The relationship between fatigue and psychiatric disorders: Evidence for the concept of neurasthenia. J Psychosom Res. 2009. doi:10.1016/j.jpsychores.2008.12.007

21. Brandes CM, Kushner SC, Tackett JL. Negative affect. Developmental Pathways to Disruptive, Impulse-Control, and Conduct Disorders. 2018. doi:10.1016/B978-0-12-811323-3.00005-5

22. Terluin B, De Boer MR, De Vet HCW. Differences in connection strength between mental symptoms might be explained by differences in variance: Reanalysis of network data did not confirm staging. PLoS One. 2016;11: 1–12. doi:10.1371/journal.pone.0155205

23. McGorry P. Transition to adulthood: The critical period for pre-emptive, disease-modifying care for schizophrenia and related disorders. Schizophr Bull. 2011. doi:10.1093/schbul/sbr027

24. Paus T, Keshavan M, Giedd JN. Why do many psychiatric disorders emerge during adolescence? Nature Reviews Neuroscience. 2008. doi:10.1038/nrn2513

25. Kessler RC, McLaughlin KA, Green JG, Gruber MJ, Sampson NA, Zaslavsky AM, et al. Childhood adversities and adult psychopathology in the WHO World Mental Health Surveys. Br J Psychiatry. 2010;197: 378–85. doi:10.1192/bjp.bp.110.080499

26. Jorm AF, Dear KBG, Rodgers B, Christensen H. Interaction between mother’s and father’s affection as a risk factor for anxiety and depression symptoms - Evidence for increased risk in adults who rate their father as having been more affectionate than their mother. Soc Psychiatry Psychiatr Epidemiol. 2003. doi:10.1007/s00127-003-0620-9

27. Costello EJ, Copeland W, Angold A. Trends in psychopathology across the adolescent years: What changes when children become adolescents, and when adolescents become adults? J Child Psychol Psychiatry Allied Discip. 2011;52: 1015–1025. doi:10.1111/j.1469-7610.2011.02446.x

28. Arrindell W, Ettema H, Groenman N, Brook F, Janssen I, Slaets J, et al. Further Dutch experiences with the Symptom Checklist-90 revised. Psycholoog. 2003. 

29. McGorry P, Nelson B. Why we need a transdiagnostic staging approach to emerging psychopathology, early diagnosis, and treatment. JAMA Psychiatry. 2016. doi:10.1001/jamapsychiatry.2015.2868

30. McGorry PD, Hartmann JA, Spooner R, Nelson B. Beyond the “at risk mental state” concept: transitioning to transdiagnostic psychiatry. World Psychiatry. 2018. doi:10.1002/wps.20514

---

## [Decision Letter · Decision Letter 1]

8 Feb 2021

Network dynamics of momentary affect states and future course of psychopathology in adolescents

PONE-D-20-11940R1

Dear Dr. Kuranova,

We’re pleased to inform you that your manuscript has been judged scientifically suitable for publication and will be formally accepted for publication once it meets all outstanding technical requirements.

Kind regards,

Geilson Lima Santana, M.D., Ph.D.

Academic Editor

PLOS ONE

Additional Editor Comments (optional):

Reviewers' comments:

Reviewer's Responses to Questions

**Comments to the Author**

1. If the authors have adequately addressed your comments raised in a previous round of review and you feel that this manuscript is now acceptable for publication, you may indicate that here to bypass the “Comments to the Author” section, enter your conflict of interest statement in the “Confidential to Editor” section, and submit your "Accept" recommendation.

Reviewer #1: All comments have been addressed

2. Is the manuscript technically sound, and do the data support the conclusions?

Reviewer #1: Yes

3. Has the statistical analysis been performed appropriately and rigorously? 

Reviewer #1: Yes

4. Have the authors made all data underlying the findings in their manuscript fully available?

Reviewer #1: Yes

5. Is the manuscript presented in an intelligible fashion and written in standard English?

Reviewer #1: Yes

6. Review Comments to the Author

Reviewer #1: COMMENTS: Since the comments made on earlier draft by me (and hopefully by other respected reviewers also) are attended positively/adequately, I am satisfied and the manuscript is improved a lot. I recommend acceptance.

7. PLOS authors have the option to publish the peer review history of their article (what does this mean?). If published, this will include your full peer review and any attached files.

Reviewer #1: **Yes: **Dr. Sanjeev Sarmukaddam

---

## [Editor Report · Acceptance letter]

9 Feb 2021

PONE-D-20-11940R1 

Network dynamics of momentary affect states and future course of psychopathology in adolescents 

Dear Dr. Kuranova:

I'm pleased to inform you that your manuscript has been deemed suitable for publication in PLOS ONE. Congratulations! Your manuscript is now with our production department. 

Kind regards, 

on behalf of

Dr. Geilson Lima Santana 

Academic Editor

PLOS ONE